# Wh-Interrogative Clauses in Istro-Romanian

**Ramona Cătălina Corbeanu** [1,2,*]  **and Ionuț Geană** [1,2,*]

1   Centre of Romanian Studies, University of Bucharest, 050663 Bucharest, Romania
2   'Iorgu Iordan–Alexandru Rosetti' Institute of Linguistics, 050711 Bucharest, Romania
*   Correspondence: catalina.corbeanu@unibuc.ro (R.C.C.); ionut.geana@litere.unibuc.ro (I.G.)

**Abstract:** This paper focuses on the syntax of interrogative clauses in Istro-Romanian. The aim is to determine the parametric settings for V-to-C, subject placement (SVO or VSO) and the target for constituent movement under discourse triggers. The findings indicate that Istro-Romanian preserved the parametric settings from Old Romanian, especially those that converged with the parametric settings in Croatian grammar. In particular, SVO can be explained only through inheritance, whereas VSO, lack of V-to-C and scrambling are a matter of both inheritance and convergence with Croatian.

**Keywords:** interrogative clauses; Istro-Romanian; scrambling; word order

## 1. Introduction

Istro-Romanian is a severely endangered Romance variety with unstable parametric settings. It is unclear at this point whether this instability is due to the heritage from Old Romanian, and/or to the bilingualism with Croatian and/or to the language's internal innovations that have only partially replaced the previous options. The fact is that first language acquisition must occur in a situation where Istro-Romanian provides unclear clues in the primary linguistic data, compared to stable clues in the acquisition of Croatian.

This paper provides examples of instability in Istro-Romanian grammar with respect to word order in wh-interrogative clauses, in root and subordinated contexts. The examples in (1) illustrate the type of data discussed in this paper: (1a) is a root clause, while (1b) shows the wh-interrogative under selection.

(1)   a.   Cum        kemåt        voi?                                                        (SF, 52)
           how        call.IND.2PL  you.PL
           'What do you call that?'
      b.   Voi        io          vedę       dende-i       cesta       faţó.       (TC, 132)
           FUT        I           see        where.from=is  this.F      scarf
           'I will see where this scarf is from.'

The focus is on the level of verb movement in interrogatives, the position of lexical subjects and constituent movement for discourse purposes. Comparison with Old Romanian helps us to understand how much of this instability could be inherited, considering that these parametric settings were also unstable in proto-Romanian around the time of population split, between the 10th and the 14th centuries. The impact of Croatian can then be estimated relative to this information, as either transfer (of a parametric setting that did not exist in Old Romanian) or convergence (reinforcing a parametric setting that existed in Old Romanian).

Briefly, we find that:

- There is no V-to-C in interrogative clauses in Istro-Romanian. This was an unproductive option in Old Romanian, which developed as a rule in Istro-Romanian, presumably under the influence of Croatian, where V-to-C is also lacking.
- The parametric setting for word order is unstable, as Istro-Romanian displays both genuine SVO and VSO. This is a general property of this grammar, so interrogatives

make no exception. Preference for SVO has been noticed in adjunct interrogatives. This unstable parameter is inherited from Old Romanian and has been preserved as such.

- Discourse triggers in interrogative clauses move constituents through scrambling within the TP field. This is a non-argumental position compatible with any type of constituent. The same type of scrambling was an option in Old Romanian at the time of population split and has been preserved in Istro-Romanian, presumably because scrambling is also productive in Croatian.

The research for this paper relied on two sources: field work and a corpus of texts. The field work documents the present day language, whereas the texts attest to the language spoken in the last century as well. The field work videos and transcriptions are available at https://www.vlaski-zejanski.com/, URL (accessed on 5 February 2024). The corpus of texts consists of the following: Traian Cantemir's *Texte istroromâne* (data collected during 1932–1933), Sextil Pușcariu's *Studii istroromâne*; *Texte I* (1906–1926) and Sârbu, R., V. Frățilă's *Dialectul istroromân* (1982–1996), added by V. Frățilă, Vasile, *Studii istroromâne* (published in 2016, but based on older material). The latter collections are also posted on the web site above. For convenience, the examples provided in this paper indicate their source in brackets only if they come from the corpus. The examples taken from the field work transcriptions are left with the generic specification (VZ).

## 2. Assessment Criteria

This paper develops an empirical analysis, aiming to sort out the relevant data rather than to demonstrate the implications of Istro-Romanian grammar for the linguistic theory. Thus, we assess the parametric settings on the basis of information and theoretical tools proposed in current formal studies without debating them.

For the position of wh-phrases, we adopt the cartographic analysis in Rizzi (1997), namely that the CP field is split as in (2), and that interrogative wh-phrases merge in Spec-FocP in this field (either base generated or moved there).

(2)    Force > TopP > FocP > FinP

The Istro-Romanian complementizer *che* 'that' is in Force when it heads declarative clauses, or in Fin when it cooccurs with wh-phrases (Corbeanu and Hill 2024). The hierarchy in (1) also allows us to assess the level of constituent movement for discourse purposes, that is, to TopP when higher than the wh-phrase, or inside TP when lower. This hierarchy was shown to hold for Old Romanian (Alboiu and Hill 2016; Nicolae 2019).

For the lower fields of the clause structure, we adopt the version of Cinque's hierarchy in (3).[1] NegP indicates the border between CP and TP. TP is the field associated with tense and agreement between verbs and its arguments (subjects and objects), while VoiceP is associated with the encoding of the thematic roles that introduce the subject (the external argument). On the other hand, vP is the field that encodes the internal arguments (direct and indirect objects). ModP and AspP are fields that encode various modal and aspectual features, and could therefore be split (which is not relevant to our inquiry).

(3)    (FinP) > NegP > TP > ModP > AspP > VoiceP > vP

In many Romance languages, the negation is a free morpheme that occurs at the border of CP and TP (Zanuttini 1997), as in (3). That was shown to be the case with Old Romanian (Alboiu and Hill 2016; Nicolae 2019), which contrasts with Balkan Slavic, including Serbo-Croatian, where negation can be a clitic on C(=Fin) and thus surface on imperative verbs in Fin (mutatis mutandi in Mišeska-Tomić 1999; Pancheva 2005). This allows us to decide on the level of verb movement and constituent movement in Istro-Romanian.

The position of lexical subjects indicates whether the language has a SVO or VSO setting. Genuine SVO means that subjects are located in Spec-TP (e.g., lower than negation but higher than simple verb forms in T), whereas derivative SVO means that subjects are in Spec-TopP, higher than negation and wh-phrases. On the other hand, genuine VSO

means that the subject surfaces in Spec-VoiceP and that Spec-TP is lost as an argumental position. Typologically, Romance languages (including proto-Romanian) are SVO, that is, their subjects can or must be licensed in Spec-TP. On the other hand, Balkan languages (including Old Romanian) are VSO, that is, the subjects are licensed in Spec-VoiceP, whereas Spec-TP is lost as an argumental position for subjects, so it can be used as a non-argumental position for discourse motivated movement of constituents (Alboiu 2002).

Finally, the level of verb movement will be assessed in relation to post-verbal subjects, auxiliary verbs, direct objects and negation. The aim is to determine whether wh-movement triggers V-to-Fin, and if that is not the case, whether the verb remains in the vP field or moves to the inflectional field (TP/AspP).

For Istro-Romanian, some of these settings have been identified for declarative root clauses and infinitive complements in previous studies, and we adopt them as such. In particular, Dragomirescu and Nicolae (2018b) show that verbs may move or not move out of vP and direct objects may occur either post-verbal (in vP) or scrambled (above vP), while clitic pronouns are either proclitics (in TP) or enclitics (in or close to vP). The question we raise is whether these options are more stable in an environment that requires wh-movement.

### 3. V-to-Fin?

When it comes to interrogative clauses, the main issue is whether wh-movement also entails V-to-Fin. Rizzi (1996) points out that V-to-C, which characterizes most modern languages of the Germanic phylum in all types of (root) clauses, is not systematic in modern Romance languages; hence the term *residual V2*, which is limited (but not cross-linguistically obligatory) to interrogative and/or relative clauses. Daco-Romanian can be included in this classification as it displays obligatory adjacency between the wh-phrase and the verb in interrogatives (hence, V-to-C; see also Motapanyane 1989, p. 97).[2]

Verb movement in Istro-Romanian wh-interrogatives will be assessed on the basis of the following criteria:

(i)   Doubly filled COMP. This is very productive in the language by the use of *che* 'that' in addition to the wh-phrase. In root clauses, *che* 'that' follows the wh-phrase, as in (4a), whereas in selected interrogatives, it precedes the wh-phrase, as in (4b).

(4) a. Če      **che**    n-åi           mes                                    våčile      čere?      (SF, 87)
       what    that    NEG= have.2SG   gone                                   cows.the    request.INF
       'Why didn't you go ask for those cows?'
    b. Prevtu   ganę    **che**    iuva ľ-e                    pucșa.                    (TC, 66)
       priest.the said   that     where he.CL.DAT-POS.3SG=is   rifle.the
       'The priest asked where his rifle was.'

Considering the hierarchy in (2), the wh-phrase is in FocP, whereas *che* 'that' may be either in Fin, in (4a), or in Force in (4b) (spelling out the complement status of the clause). Doubly filled COMP occurs in other environments as well in Istro-Romanian (e.g., subjunctive clauses; Corbeanu and Hill 2024) and indicates whether Fin is available for *che* insertion, that is, when there is no V-to-Fin or another competing element in Fin.

(ii)  Non-clitic AUX. Old Romanian auxiliaries were unbound morphemes up to around the 15th century, with tail end occurrences in 16th century texts (Nicolae 2019). By the 16th century, the auxiliaries were treated as clitics. Istro-Romanian preserved the non-clitic AUX, as shown by the interpolation of material in (5a) and the possibility of verb ellipsis in (5b). Theoretically, this property provides the option for AUX-to-Fin under certain triggers, including interrogative C features. The data should indicate whether such movement is or is not implemented in Istro-Romanian wh-interrogatives.

(5) a. Ie       lucråt-a       cum    ľ-a              [orălu]         **zis**.                (TC, 29)
       he       worked=has     how    him.CL.DAT=has   vulture.the     said
       'He acted how the vulture had told him.'
    b. Jo nu    štivu          juva    ča      mes-**a**      **fini,**   juva    n-**a**.        (VZ)
       I NEG    know           where   that    gone=has    finish   where   NEG=has
       'I don't know what has or has not come of that.'

(iii)   Clitic pronouns. The location of clitic pronouns in Istro-Romanian is very unstable, as they may surface at various locations: rarely very high, as in (6a), above the negation, hence at C; by default it is high, above AUX, as in (6b), hence at T and lower than AUX, presumably lower than AspP, as in (6c), which some studies locate in vP (Dragomirescu and Nicolae 2020). Despite this instability, there are instances where clitics may provide some clues. In particular, proclitics on AUX indicate that AUX did not raise.

(6) a.  tu-**m**          **ń**-ai                    fost        poredņe      (SP, 29)
         you=CL.DAT.1SG    not=AUX.PERF.2SG           be.PPLE     bad
         'you weren't mean to me'
     b.  **Ĺ**-å          mes                         cľemå.                   (TC, 52)
         CL.ACC.M.3SG=AUX.PERF.3SG   go.PPLE          call.INF
         'He went to call him.'
     c.  și              pus-a                        **vo**      ăn cådẹre   cuhẹi   (TC, 81)
         and             put.PPLE=AUX.PERF.3SG        CL.ACC.F.3SG  in bucket  boil.INF
         'and he put it in the bucket to boil'

These criteria will be separately applied to simple and complex verb forms in wh-interrogatives, since previous studies signaled a difference in the level of verb movement in general for the two inflectional forms (Dragomirescu and Nicolae 2018b). For example, in declarative clauses, simple verb forms were shown to undergo V-to-T in alternation with staying low in the TP field, so they support proclitics and stay lower than the negation, as seen with *pote* 'can' in (7a). With complex verb forms, the unbound AUX in T blocks V-to-T, so the main verb either remains lower, as was shown in (5a) and further in (7b), or it undergoes Long Head Movement (LHM; Lema and Rivero 1989), that is, V-to-Fin across AUX, as in (7c).

(7) a.  Acmó            nu        se                 **póte**        cåsa cumparå   (SF, 119)   V-to-T
         now             NEG       CL.REFL.PASS       can.PRES.3SG    house buy.INF
         'Now the house cannot be bought.'
     b.  Bire. **Ŭam**   **slujit**   un cesar                                      (TC, 10)    -LHM
         good AUX.PERF.1SG   serve.PPLE   an emperor
         'Good. I served an emperor.'
     c.  **Pus-ľ-a**       mărle.                                                   (TC, 8)     +LHM
         put.PPLE=CL.DAT.3SG=AUX.PERF.3SG   hands.the
         'He put his hands on him.'

While these properties are unexceptional cross-linguistically, the challenge with Istro-Romanian is that they do not apply systematically. For example, there is no way to determine what makes the speaker opt or not opt for LHM.

### 3.1. Simple Verb Forms

The data in our corpus indicate that Istro-Romanian systematically lacks V-to-Fin in all the environments with wh-phrases and simple verb forms. First, root clauses display constituents between the wh-phrase in FocP and the simple verb form, as in (8a), indicating lack of V-to-Fin. Thus, Fin may be occupied by *che* 'that', yielding doubly filled COMP, as in (4a) and (8b). The negation in (4b) provides further confirmation for the absence of V-to-Fin, as it would block such movement.

- Root interrogatives

(8) a.  Če              [io]       știu?                    (SF, 138)
         what            I          know.IND.1SG
         'What do I know?'
     b.  Če              ke         lucråm?                  (SF, 105)
         what            that       work.IND.1PL
         What are we doing?'

The same lack of V-to-Fin is systematic in subordinate clauses, where constituents intervene between the wh-phrase and the verb, as shown in (9) and (10).

- Adjunct interrogatives

(9)      Saka    zi,      saka    sẹra     kând    [god]     je          vrur.     (VZ)
         every   day      every   evening  when    around    be.IND.3SG  anyone
         'Every day, every evening [it happens], when there are people around.'

- Selected interrogatives

(10)  a.  | L-av | | ăntrebåt | iuvẹ | [ie] | mẹre. | (SI, 15) |
| --- | --- | --- | --- | --- | --- | --- |
| him.CL.3SG.M.ACC=has | | ask.PPLE | where | he | go.IND.3SG | |

'He asked him where he was going'

b.  | Acmó | voi | spúre | cúmo | [Bojíču] | dočkẹim. | (SF, 71) |
| --- | --- | --- | --- | --- | --- | --- |
| now | will.1SG | say | how | Christmas | wait.IND.1PL | |

'Now I will say how we prepare/wait for Christmas'

Comparatively, Old Romanian texts from the 16th century show that wh-phrase-verb adjacency (presumably V-to-Fin) is the default configuration in root interrogative clauses, although examples with an intervening adverb are also found, albeit rarely (Pană Dindelegan 2016, pp. 580–3). In the subsequent centuries, V-to-Fin becomes the rule with interrogatives and stays so in Daco-Romanian. On the other hand, Croatian wh-interrogatives display a variety of constituents that can intervene between wh-phrases and the verb (Brown and Alt 2004), as shown in (11).

(11)  | S | kim | [Marija] | radi? | (from Brown and Alt 2004, p. 65) |
| --- | --- | --- | --- | --- |
| with | whom | Maria | works | |

'With whom does Maria work?'

The inference is that Istro-Romanian has inherited an unstable parameter from Old Romanian (either with or without V-to-Fin) which became stabilized as no V-to-Fin under language contact with Croatian.

*3.2. Complex Verb Forms*

Complex verb forms in Istro-Romanian consist of an auxiliary and a past participle or infinitive main verb. These auxiliaries are free morphemes base generated in T, so, by default, they host proclitic pronouns and linearly follow the negation (Dragomirescu and Nicolae 2018a, 2020). Two word orders are found with complex verb forms in wh-interrogatives in Istro-Romanian: one in which the auxiliary remains in T and the main verb is lower and one in which the main verb surfaces above the auxiliary in T.

Let us look, first, at the order in which AUX is higher than the main verb. In the root clause in (12a), this word order allows for proclitics on AUX, indicating lack of AUX-to-Fin. In (12b), the negation is present, blocking any head movement to Fin.

(12)  a.  | Ce | ti-ŭam | io | facut? | (TC, 8) |
| --- | --- | --- | --- | --- |
| what | you.CL.2SG.DAT=have-1 | I | done | |

'What have I done to you?'

b.  | Ce | n-a | ie | popit? | (TC, 11) |
| --- | --- | --- | --- | --- |
| what | NEG-has | he | drunk | |

'What hasn't he drunk?'

The same word order appears in adjunct and selected wh-interrogatives, as in (13) and (14), respectively.

(13)  | Ie | lucrât-a | **cum** | l'-a | orălu | zis. | (TC, 29) |
| --- | --- | --- | --- | --- | --- | --- |
| he | worked=has | how | him.CL.DAT=has | vulture.the | said | |

'He acted as the vulture had told him.'

(14)  a.  | Ŭǎi | vezut | **cum** | te-ŭǎm | io | privarit? | (TC, 83) |
| --- | --- | --- | --- | --- | --- | --- |
| have-2 | seen | how | you.SG.CL.ACC=have-1 | I | tricked | |

'Did you see how I tricked you?'

b.  | Acmó | misles | cumo | mi-åu | åńi | trecut. | (SF, 74) |
| --- | --- | --- | --- | --- | --- | --- |
| Now | think.IND.1 | how | CL.1SG.DAT-POSS=have-3 | years | passed | |

'Now I am thinking about how my years have passed.'

This approach entails that the order in which AUX is followed by clitics, as in (15), may indicate AUX-to-Fin. However, as mentioned in the previous section, clitic pronouns may be located very low in the clause hierarchy, not only at T.

(15)  | Io | n-oi | ști | **iuva** | voi | ve | duce | obedu. | (TC, 85) |
| --- | --- | --- | --- | --- | --- | --- | --- | --- |
| I | NEG=will-1 | know | where | will-1 | you.SG.CL.ACC | bring | lunch | |

'I won't know where to bring your lunch.'

The disambiguating test for the order in (15) would be an order in which the AUX-clitic is preceded by negation. We could not find that order, which makes AUX-to-Fin a reasonable assumption. If that is the case, we can conclude that Istro-Romanian wh-

interrogatives have no AUX-to-Fin as the default rule, although exceptions may also be found.

A completely different word order in these constructions involves material and the main verb above AUX. This is equally found in root and subordinate wh-interrogatives, as in (16)–(18).

- Root clause

(16)  a.  Cum        [vo                        scapuľeit]-a?                                    (TC, 91)
          how        her.CL.3SG.F.ACC           saved=has
          'How did he save her?'
      b.  Cire       [cåșu                       și                     hlebu           poidit]-a?   (TC, 62)
          who        cheese.the                 and                    bread.the       ate=has
          'Who ate the cheese and the bread?'

- Adjunct

(17)  a.  Cănd       [casunu rescľis]-a,         Madalena               fost-a    ăn        casun    (TC, 97)
          when       box.the opened=has         Madalena               been=has  in        box
          'When he opened the box, Madalena was inside that box.'
      b.  Mes-a      cåșe,                       cătra cesåru,          iuva      [din       måre
          went=has   house                      at emperor.the         where     from       big
          jålostăn   verit]-a                    mare                   veseľe.                       (TC, 15)
          pain       come.PPLE=has              big                    joy.
          'He went to the emperor's house, where out of great pain came great joy.'

- Selected

(18) a.  Nu        știvu       cumo       [d'atunce  obârnit]-av      se       de     sus.        (SF, 52)
         NEG       know-1      how        of=then    came=have        REFL.3PL from   above
         'I don't know how then they came back from uphill.'
     b.  Jo nu štivu    juva       [ča     mes]-a       fini,           juva              n-a.       (VZ)
         I NEG know-1   where      that    gone=has     end.up.INF      where             NEG=has
         'I don't know what has or has not come of that.'

A similar order may occur in declaratives, but only with clitics preceding the main verb, as in (19). However, in wh-interrogatives the clitics can alternate with substantive constituents, as seen above.

(19)      [Vo                      ve'zut]-a          nuškarlji.          (AD, 42)
          her.CL.3SG.F.ACC         saw=has            someone
          'Someone saw her.'

Neither the Old Romanian texts nor the Croatian speakers we asked attest to the word order in (16b)–(18), which indicates a language internal innovation. One may argue that this word order involves LHM, since LHM has been shown to take place in declarative clauses, as mentioned for (7c) and repeated as in (20a). However, LHM does not carry the clitics, so the proclitics on the main verb, as in (16a), are unexpected. Clitics should remain on AUX, as in (20a), or lower than AUX, as in (6c) and repeated as in (20b).

(20) a.  Pus-ľ-a                                 mărle.                                          (TC, 8)
         put.PPLE=CL.DAT.3SG=AUX.PERF.3SG       hands.the
         'He put his hands on him.'
     b.  și                                      pus-a                    **vo**           ăn cădęre   cuhęi    (TC, 81)
         and                                     put.PPLE=AUX.PERF.3SG    CL.ACC.F.3.SG   in bucket    boil.INF
         'and he put it in the bucket to boil'

One may argue that the clitics preceding the moved verb are base generated at C, as in Balkan Slavic, and thus may procliticize on the verb in Fin. However, the substantive constituents that may also precede the moved verb cannot do that, so the construction fails to qualify for either Wackernagel law or V2 in the presence of XPs.

We suggest that this word order is a language internal innovation that exploits the possibility of phrasal vs. head movement. More precisely, the features of Fin that get checked through head-to-head movement (i.e., LHM) may also be checked through phrasal movement of AspP to Spec-FinP. AspP must necessarily contain the verb in order to qualify as a phase, in the terms of Bošković (2014) (i.e., the functional projection of a lexical category). In other words, what moves to Spec-FinP in (16)–(18) is the entire clause structure from AspP down, after the vP has been vacated of constituents. This is shown in (21).

(21) a.  [ForceP Force [qu][FocP WH-op [FinP Fin [TP Aux [AspP XP/CL Asp [VoiceP Voice-verb[vP <XP> <verb>]]]]]]
     b.  [ForceP Force [qu][FocP WH-op [FinP **AspP** Fin [TP Aux <[AspP XP/CL Asp [VoiceP Voice-verb [vP <XP> <verb>]]]]]]

In (21a), the clitic or XP constituent move to the lower part of the inflectional field. The latter is not different from the scrambling operation discussed in Section 5 below. In (21b), the entire AspP (which includes VoiceP/vP) moves to Spec-FinP because of a Fin probe.

The aim of this section was to determine whether Istro-Romanian wh-interrogatives display V-to-Fin. Technically, they do not, if V-to-Fin means (residual) V2, through head-to-head movement. If that were the case, the data should have provided proof of AUX-to-Fin or LHM, as a condition for grammaticality. However, an alternative to V-to-Fin is available, in the guise of phrasal movement to Spec-FinP, which must be justified through discourse triggers, since it is not a condition for grammatical output.

### 4. SVO or VSO?

Istro-Romanian has a generalized null subject parametric setting. When it comes to lexical subjects, they surface in either a pre-verbal or post-verbal position, in root and subordinate clauses, and in declarative and in interrogative clauses, as shown in (22)–(24). This is unexceptional for Balkan languages.

- Root clauses

(22) a.
| Cum | o | **voi** | zičeț? | (SF, 64) |
|---|---|---|---|---|
| how | her .CL.3SG.F.DAT | you.PL | say.IND.3 | |

'What do you call it?'

b.
| Ma | cum | ke | nu | mi-e | **niș**? | (SF, 99) |
|---|---|---|---|---|---|---|
| but | how | that | NEG | me.CL.1SG.DAT=is | nothing | |

'But how come there's nothing wrong with me?'

- Adjunct clauses

(23) a.
| Cănd | **Mario** verire, | io | l-oi | zaino ucide | (TC, 94) |
|---|---|---|---|---|---|
| when | Mario come.FUT.3.SG | I | him.CL.3SG.M.ACC=will-1 | always kill.INF | |

'When Mario comes, I will kill him right away.'

b.
| Verįt-a | įuvẹ | s-a | copt | **pǎra** | ǎn coptór | (SI, 42) |
|---|---|---|---|---|---|---|
| came=has | where | REFL=has | baked | bread.the | in oven | |

'He came where bread was baked in the oven.'

- Selected clauses

(24) a.
| N-åm | caută̊t | cumo | **focu** | årde. | (SF, 302) |
|---|---|---|---|---|---|
| NEG=have-1 | searched | how | fire.the | burn.IND.3 | |

'I wasn't paying attention to how the fire was burning.'

b.
| Utåt-a | če | ľ-a | zis | **guårdiĭa**. | (TC,12) |
|---|---|---|---|---|---|
| forgot=has | what | him.CL.3SG.DAT=has | told | guard.the | |

'He forgot what the guard had told him.'

In Balkan languages, SVO arises from movement of subjects to TopP or FocP, so they surface above wh-phrases. This was also the default order for Old Romanian, as shown in (25).

(25)
| [Neamul | Țării | Moldovei] | de unde | se | tărăgănează? | (MC, 6) |
|---|---|---|---|---|---|---|
| people.the | country.GEN | Moldova.GEN | from where | REFL | originate.IND.3 | |

'Where do the ancestors of Moldova originate from?'

However, in Istro-Romanian, pre-verbal subjects occur lower than the wh-phrase, as in (22a), (23a) and (24a). Hence, these subjects are in Spec-TP, which is a property of Romance languages. Old Romanian texts from the 16th century also provide some examples of pre-verbal subjects in Spec-TP, as a tail end of a Romance parametric setting which was being replaced with the Balkan one (Alboiu and Hill 2017; Nicolae 2019). It is reasonable to assume that, at the time of population split, the Romance SVO setting was the productive option, and it has been preserved as such in Istro-Romanian.

Post-verbal subjects, as in (22b), (23b) and (24b), attest to the concurrent option for Balkan VSO. These post-verbal subjects remain in situ in Spec-VoiceP, as opposed to being right dislocated. This must be the case because (i) these are unmodified nouns, so 'heavy NP-movement' to the right is out of the question; and (ii) they may be bare quantifiers, such as *niș* 'nothing' in (22b). Bare quantifiers must remain in an argument position in syntax (vs. being dislocated) to allow for quantifier raising at LF (Cinque 1990).

In sum, Istro-Romanian has an unstable parametric setting for subject placement: they may be in Spec-TP, yielding SVO, or in Spec-VoiceP, yielding VSO. Although Balkan languages also display a mix of SVO and VSO in the word order, their SVO is discourse

driven and involves Spec-TopP, whereas the SVO with the subject in Spec-TP, as in Istro-Romanian, is a licensing configuration.

## 5. Scrambling

Istro-Romanian uses the field above VoiceP for movement of constituents under discourse triggers. This is expected with languages that have non-clitic auxiliaries. However, in Istro-Romanian this may also be the case with simple verb forms, and that can equally be found in interrogatives, as in (26a), and in declaratives, as in (26b).

(26) a.
| E | cúmo | io | [trei | zile] | [såmo | åpa] | be? | (SF, 124) |
|---|------|-----|-------|-------|-------|------|-----|-----------|
| and | how | I | three | days | only | water | drink-1 | |

'And how am I to drink only water for three days?'

b.
| Nu | se | [nícad] | iâve. | (SF, 80) |
|----|-----|---------|-------|----------|
| NEG | CL.REFL.3SG | never | showed | |

'He has never showed himself'

In both (26a,b) the scrambled material surfaces between Spec-TP (with the subject *io* 'I' in (26a)) or T (with the reflexive clitic *se* following the negation in (26b)) and the inflected verb in Voice. In this space, there are two positions available for scrambling, as visible in (26a) with two different types of XPs.

Crucially, these are non-argumental positions, since PPs or AdvPs may move there. Thus, scrambling in Istro-Romanian does not follow from a parametric setting for OV. In fact, the scrambled material may also alternate with its post-verbal placement, especially with simple verb forms, with no change in meaning (Maiden 2016, p. 118). The pre-verbal placement seems to be a question of pragmatics, under discourse triggers (e.g., discourse continuity or new information).

Movement for discourse purposes in Old Romanian also involved scrambling in the presence of complex verb forms at the time when the auxiliaries were non-clitic, as the main verb remained low. As the auxiliaries became clitic, movement to the CP field developed (i.e., TopP or FocP) and scrambling was lost (Alboiu and Hill 2017). Thus, it is unsurprising that Istro-Romanian preserved scrambling since it has also preserved non-clitic auxiliaries. In addition, productivity of scrambling in Croatian encouraged the maintenance of this parametric setting and its extension to contexts with simple verb forms, as in (26).

The following examples illustrate scrambling in wh-interrogatives in root, adjunct and selected clauses:

- Root clauses

(27) a.
| Če, | če | s-av | mije | fakut? | (VZ) |
|-----|-----|------|------|--------|------|
| what | what | REFL.ACC=has | me.DAT | done | |

'What has happened to me?'

b.
| Ši | kând | va | mije | fratele | veri? | (VZ) |
|----|------|-----|------|---------|-------|------|
| and | when | will-3 | me.DAT | brother.the | come | |

'And when will my brother come to me?'

c.
| Ma | ce-ŭåi, | dråcu, | fåcut? | (TC, 46) |
|----|---------|--------|--------|----------|
| but | what=have-2 | devil.the.VOC | done | |

'But what have you done, devil?'

d.
| Če | reş | ânca | ziče? | (SF, 83) |
|----|-----|------|-------|----------|
| what | would-1 | still | say | |

'What else should I say?'

- Adjunct clauses

(28)
| Io-m | fost | iuva | n-a | ieľ | nicad fos. | (TC, 28) |
|------|------|------|-----|-----|------------|----------|
| I=have-1 | been | here | NEG=have | they | never been | |

'I've been where they have never been.'

- Selected clauses

(29)
| Știři | iuva | ver | tu | cmoce | mẹre? | (TC, 28) |
|-------|------|-----|-----|-------|-------|----------|
| know-2.SG | where | will | you.SG | now | go | |

'Do you know where you will go now?'

These data show that AUX follows the negation (e.g., in (28)) and carries proclitics, so it stays in T in the presence of scrambling. As the main verb stays in Voice (Dragomirescu and Nicolae 2018b), the scrambling positions may involve the vP periphery identified in Belletti (2008) as allowing for projections that accommodate constituents with certain topic

or focus reading. Thus, these positions are higher than subjects in situ (in Spec-VoiceP), as confirmed by the word order in (27b).

That being said, subjects may also undergo scrambling, in which case they surface between AUX and another pre-verbal constituent, as in (28) and (29). The location of subjects between AUX and the low verb form is very productive, as further shown in (30)–(32). However, it is hard to decide on their exact location in the absence of concurrent clues. They may be either in situ or scrambled.

Root clauses

| (30) | a. | Juva | ac | va | voj | igrejt? | (VZ) |
| | | where | have-2.PL | REFL | you.PL | played | |
| | | 'Where did you play?' | | | | | |
| | b. | Če-l | voj | jo | dǎ? (VZ) | | |
| | | what=her.CL.ACC | will-1 | I | give | | |
| | | 'What will I give her?' | | | | | |
| | c. | Ce | ren | noi | face | cmo? | (TC, 60) |
| | | what | would-1.PL | we | do | now | |
| | | 'What shall we do now?' | | | | | |

Adjunct

| (31) | Ie lucrǎt-a | cum | ľ-a | orǎlu | zis. | (TC, 29) |
| | he worked=has | how | him.CL.ACC=has | vulture.the | told | |
| | 'He did as the vulture had told him.' | | | | | |

Selected

| (32) | a. | Ŭǎi | vezut | cum | te-ǔǎm | io | privarit? | (TC, 83) | |
| | | have-2SG | seen | how | you.CL.SG=have-1 | I | tricked | | |
| | | 'Did you see how I tricked you?' | | | | | | | |
| | b. | Nu . . . | poc | aflǎ | cum | mi-av | mǎia | zis. | (TC, 113) |
| | | NEG | can-1 | find.out | how | me.CL.DAT=has | mother.the | told | |
| | | 'I cannot find out what my mother called me.' | | | | | | | |

In sum, Istro-Romanian displays scrambling as a means of moving constituents to non-argumental positions. This is a discourse motivated operation, as opposed to scrambling for licensing in a verb final configuration. Any type of constituent may undergo scrambling in Istro-Romanian, including subjects. The possibility for scrambling arises from the nature of verb movement, which may remain in a low position.

## 6. Conclusions

This paper focused on one type of clause in Istro-Romanian, namely wh-interrogatives, in order to determine the parametric settings for verb movement, subject position and constituent movement under discourse triggers. The overall observation is that these parametric settings are unstable. There is no V-to-Fin, but there is phrasal movement to Spec-FinP. There is genuine SVO, but also genuine VSO. There is interpolation between the items in T and the verb in Voice, but this interpolation may arise either from scrambling or from subjects in situ. This variability in the syntax of wh-interrogatives matches similar observations for other Istro-Romanian clause types (e.g., declaratives in Dragomirescu and Nicolae 2021; subjunctives in Corbeanu and Hill 2024) where the parametric settings are also volatile.

**Author Contributions:** R.C.C. and I.G. have jointly worked for this article. All authors have read and agreed to the published version of the manuscript.

**Funding:** This research received no external funding.

**Institutional Review Board Statement:** Not applicable.

**Informed Consent Statement:** Not applicable.

**Data Availability Statement:** The original contributions presented in the study are included in the article, further inquiries can be directed to the corresponding authors.

**Conflicts of Interest:** The authors declare no conflict of interest.

## Notes

1   Cinque (1999) also has a lower TP field that is not included in (3). The lower TP is important for the assessment of the placement of clitic pronouns (see Corbeanu and Hill 2024). Clitic pronouns are not the focus of discussion in this paper, so we did not use a finer-grained structure for them.

2   Residual V2 was shown to arise only when there is wh-movement and operator-variable chains. This excludes 'why' questions, where the wh-phrase is base generated in CP and does not head an operator-variable chain.

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
