# Peer review of "Wh-Interrogative Clauses in Istro-Romanian"

_languages, doi:10.3390/languages9020064_

Round 1

Reviewer 1 Report

Comments and Suggestions for Authors

The article is well written, has clear theoretical objectives, supported by well chosed data.

I think the author shoud introduce 2-3 relavant examples with IR interrogatives at the beginning of the paper, in order to make the question clearer for the reader, since the first data are on page 3.

I will suggest, if possible, to explore more the consequences of Istro-Romanian-Croatian contact in syntax: for example, Hickey, Raymond (ed.), 2010, The Handbook of language contact, Wiley-Blackwell, for the concept of convergence, or Ross, Malcolm, 2019, ‘Syntax and Contact induced language change’, in: Anthony P. Grant (ed.), The Oxford Handbook of Language Contact, Oxford, Oxford University Press, 123-154, for the general consequences of language contact in syntax.

 Minor suggestions:

- please alingn the glosses (3b, 4b, 5a,b,c, 6a,b,c, 7a, 8, 9a,b, 12, 13b, 14, 16b, 17b, 18, 19b, 21, 22, 25, 26b,c,d, 27, 28)

- please mark the translations in '' everywhere (3b)

- ex. 15a, gloss: how, instead of How

Author Response

Thank you for your observations! We introduced, at the beginning of the paper, 2 relevant examples with IR interrogatives (in a root clause and in the wh-interrogative under selection) in order to make the question clearer for the reader. About the minor suggestions, we aligned the glosses and marked the translation mentioned.

Reviewer 2 Report

Comments and Suggestions for Authors

The topic of the article is interesting and novel, and the analysis is clear and accurate. The data analysis is based on an up-to-date bibliography, and it is conducted within a relevant theoretical framework.

Author Response

Thank you for your review and for helping us improve the quality of our paper! We are glad for your interest in our topic!